# MicroRNAs with Multiple Targets of Immune Checkpoints, as a Potential Sensitizer for Immune Checkpoint Inhibitors in Breast Cancer Treatment

**DOI:** 10.3390/cancers15030824

**Published:** 2023-01-29

**Authors:** Huiling Zhou, Wentao Jia, Lingeng Lu, Rui Han

**Affiliations:** 1Department of Chinese Medicine Oncology, The First Affiliated Hospital of Naval Medical University, Shanghai 200433, China; 2Department of Chinese Medicine, Naval Medical University, Shanghai 200433, China; 3Department of Oncology, Yueyang Hospital of Integrated Traditional Chinese and Western Medicine, Shanghai 200437, China; 4Department of Chronic Disease Epidemiology, Yale School of Public Health, New Haven, CT 06520-8034, USA; 5School of Medicine, Center for Biomedical Data Science, New Haven, CT 06520-8034, USA; 6Yale Cancer Center, Yale University, New Haven, CT 06520-8034, USA

**Keywords:** microRNA, multiple immune checkpoints blockade, sensitizer, breast cancer, TMTME

## Abstract

**Simple Summary:**

Based on the latest research progresses, application of immune checkpoint inhibitors (ICIs) has shown promise in treating breast cancer. Moreover, novel ICIs based combined therapy has been sought to further enhance the curative effect. This review brings up a whole new conception of combined strategy by adding miRNA therapy into immune checkpoint blockade (ICB), based on the fact that miRNAs targeting multiple immune checkpoint molecules are believed to enhance the efficacy of ICB by mimicking combination therapy. Potential miRNAs have been summarized in this study. We also discussed the potential side-effects and solutions of applying such method. To thoroughly evaluate the role of miRNAs with multiple immune checkpoint molecules to act as a novel additive therapy for ICB in cancer treatment in future study, may further improve the clinical benefit of cancer immunotherapy.

**Abstract:**

Breast cancer is the most common cancer type and the leading cause of cancer-associated mortality in women worldwide. In recent years, immune checkpoint inhibitors (ICIs) have made significant progress in the treatment of breast cancer, yet there are still a considerable number of patients who are unable to gain lasting and ideal clinical benefits by immunotherapy alone, which leads to the development of a combination regimen as a novel research hotspot. Furthermore, one miRNA can target several checkpoint molecules, mimicking the therapeutic effect of a combined immune checkpoint blockade (ICB), which means that the miRNA therapy has been considered to increase the efficiency of ICIs. In this review, we summarized potential miRNA therapeutics candidates which can affect multiple targets of immune checkpoints in breast cancer with more therapeutic potential, and the obstacles to applying miRNA therapeutically through the analyses of the resources available from a drug target perspective. We also included the content of “too many targets for miRNA effect” (TMTME), combined with applying TargetScan database, to discuss adverse events. This review aims to ignite enthusiasm to explore the application of miRNAs with multiple targets of immune checkpoint molecules, in combination with ICIs for treating breast cancer.

## 1. Introduction

Breast cancer, as one of the most common malignant tumors in women, also accounts for a huge part of cancer-associated deaths world widely. It undoubtedly has become the top killer, severely threatening women’s health [1]. As reported by the International Agency for Research on Cancer (IARC), the disease has already surpassed lung cancer as the leading cause incidence, which made breast cancer the top cancer in 2020 [2]. Moreover, triple-negative breast cancer (TNBC), as a special subtype, can bring even worse prognosis, mortality, and a higher relapse for patients [3]. Therefore, a massive effort has been made for improving the outcome of breast cancer treatment globally. In recent years, as one of the most significant advances, the application of immune checkpoint inhibitors (ICIs) has been considered to possess a promising potential in breast cancer treatment (including TNBC) [4,5]. However, although many immune checkpoint inhibitors (ICIs) have manifested compelling clinical effectiveness for breast cancer, the stable and ideal clinical benefits were still hard to gain (such as the results from the IMpassion050 Trial) [6,7,8], which makes it a global research hotspot how to refine the combination of ICIs to enhance the therapeutic effect.

Interestingly, targeting multiple immune checkpoints has been demonstrated to be a more effective approach to activate anti-tumor immune responses than single immune checkpoint-specific monotherapy [4], which makes the combinations, such as CTLA-4 (cytotoxic T lymphocyte-associated protein-4) blocker plus PD-1/PD-L1 (programmed cell death protein-1/programmed death-ligand 1) blocker, a very promising anti-cancer therapeutic strategy, including against breast cancer [4,9]. Based on that, agents which may further improve the therapeutic effect of multiple immune checkpoints blockade are being explored [9]. For instance, T-DM1 (ado-trastuzumab emtansine) has been found to enhance the curative effect of CTLA-4/PD-1 blockades for breast cancer [9]. Moreover, the approval of cadonilimab has officially opened up a new era for multiple immune checkpoint inhibitors’ application in solid cancers, which also brings a greater opportunity for treating breast cancer [10].

MiRNAs (MicroRNAs) are a class of non-coding RNAs that plays an important role in the post-transcriptionally controlling the expression of messenger RNA (mRNA) translating into protein [11]. One single miRNA has been found to regulate nearly a hundred mRNAs (messenger RNAs); therefore, robust biological responses can be created by simultaneous stimulation on multiple gene networks [11,12,13]. Extracellular miRNAs have also been found to mediate cell–cell communication as chemical messengers [14]. MicroRNA (miRNA)-based therapeutics can be divided into miRNA mimics and inhibitors of miRNAs (antimiRs) [15]. The miRNA mimics are synthetic small RNA molecules which can target and combine the corresponding miRNA sequence for replenishing the lost miRNA expression in certain diseases [15]; meanwhile, miRNA inhibitors have been applied to block the expression of certain oncogenic microRNAs [16]. This miRNA-based therapy has been considered as a potential candidate for cancer treatment, since abnormal expressions of certain genes were found to have inevitable connection with tumorigenesis and cancer development [12,13,15].

Notably, a single miRNA has been shown to target multiple checkpoint molecules, mimicking the therapeutic effect of a combined immune checkpoint blockade (ICB) [17]. As a result, microRNA therapy combined with ICB can potentially increase the efficacy of the established mono-therapeutic approach [18]. In actuality, not only is the ORR (objective response rate) induced by ICIs highly restricted, but the number and severity of adverse effects caused by combinations of immune checkpoint inhibitors are frustrating. Consequently, it is of the utmost importance to investigate novel ways to improve the situation. MiRNAs, particularly those that can target multiple immune checkpoints simultaneously, simulating a multi-checkpoint blockade, have been deemed suitable for enhancing the efficacy of immune checkpoint inhibitors. This strategy is believed to positively modulate the immune response to the treatment and convert non-responders into responders [18]. Even though such a combination approach has not yet reached the stage of clinical trials, there is growing preclinical evidence of potential synergistic interactions between immune checkpoint blockade and microRNAs [19,20] For example, PD-L1 and CD80 expression negatively correlated with miR-424 in ovarian cancer cells. Mimetics of miR-424 are believed to increase the therapeutic efficacy of immunotherapy [21]. In the context of ICB, MiR-138, which can regulate the expression of PD-1 and CTLA-4 in gliomas, has also been deemed to possess a significant therapeutic potential [22]. The intertwined signal networks between miRNAs and immune checkpoint molecules are robust and intricate, resulting in numerous inevitable and formidable obstacles to applying such synergy [23]. Therefore, we are here to summarize potential miRNAs, which can target multiple immune checkpoints to ignite the enthusiasm of exploring the possibilities of such miRNAs combined with ICIs in breast cancer treatment. We also summarized the potential side effects of such synergism and possible solutions to preliminarily construct an overall prospect for such a novel therapeutic strategy in breast cancer treatment.

## 2. Method

The TargetScan database (https://www.targetscan.org/vert_80/, accessed on 29 November 2022) (TargetScan Release 8.0) was applied to predict the potential targets of selected miRNAs.

## 3. Results

As displayed by the analyses performed using the TargetScan platform, the number of predicted targets of selected miRNAs ranges from 59 to 7852. For details, miR-149-3p has been found to possess 7852 transcripts with binding sites, containing a total of 17186 sites. The miR-195/miR-497 has 1515 transcripts with conserved sites, including 1769 conserved sites and 687 poorly conserved sites. The miR-93 has 1385 transcripts with conserved sites, containing a total of 1647 conserved sites and 914 poorly conserved sites. As for miR-5119, the number of transcripts with sites binding is 2537, containing a total of 3078 binding sites. The number of transcripts with conserved sites of miR-340 are 1393, containing a total of 1624 conserved sites and 1722 poorly conserved sites. The miR-21 possesses 384 transcripts with conserved sites, including 414 conserved sites and 138 poorly conserved sites. For miR-138, 704 transcripts with conserved sites were predicted (including 787 conserved sites and 306 poorly conserved sites). The miR-100 has been predicted to have 59 transcripts with conserved sites, containing 60 conserved sites and 2 poorly conserved sites, and the predicted counterparts for miR-200a are 905, 1027, and 566, respectively. The miR-4443 has been predicted to possess 4481 transcripts with binding sites, containing a total of 6052 sites (Table 1).

## 4. Discussion

### 4.1. Existing and Potential Immune Checkpoints in Breast Cancer

In recent years, a great progression has been witnessed in the application of ICIs in cancer treatment, including against breast cancer [38]. For instance, it has been presented by the IMpassion130 study that the therapeutic strategy of atezolizumab (a PD-1/PD-L1 inhibitor) plus albumin paclitaxel as the first-line therapy has improved the overall survival (OS) by 7.5 months and reduced the risk of death by 33%, compared with the placebo plus albumin paclitaxel group, for advanced TNBC patients with PD-L1 positive, marking the beginning of the immunotherapy for breast cancer [39]. Consequently, selecting proper immune checkpoint as an effective therapeutic target is a fundamental and crucial aspect of immunotherapy. We now have presented some potential candidates which may act as therapeutic targets in multiple immune checkpoint targeted miRNA therapy (Table 2 and Figure 1).

#### 4.1.1. PD-1/PD-L1

PD-1, or differentiated cluster 279 (CD279), is an important immunosuppressive molecule. As a cell surface receptor and a member of the CD28 family, this molecule is expressed in lymphocytes, such as T cells, B cells, natural killer (NK) cells, and myeloid cells [59]. PD-L1 (CD274), a ligand for PD-1, is a type 1 transmembrane protein. PD-L1 can be glycosylated by N-linked glycoproteins in the endoplasmic reticulum and Golgi apparatus, and then transported to the cell membrane as a ligand for PD-1. There is evidence that PD-L1 is highly expressed in some breast cancers, and the positive rate of PD-L1 in triple-negative breast cancer can reach 20% [60,61]. In addition, it has been discovered that the binding of PD-L1 to PD-1 inhibits T cell proliferation and activation, resulting in immunosuppression. Based on this fact, inhibitors of PD-1 or PD-L1have then been developed for cancer treatment (including breast cancer), by restoring anti-cancer immunology [43]. These PD-1/PD-L1 inhibitors, such as atezolizumab and pembrolizumab, have already been approved for application in breast cancer treatment [43]. However, the therapeutic effect of PD-1/PD-L1 inhibitor monotherapy for breast cancer is limited, resulting in more attention on the multiple immune checkpoint blockade therapeutic strategy [62,63]. For instance, a phase I/II study (NCT05187338) is currently testing the triplex CTLA4/PD1/PDL1 inhibitors combination therapy for advanced solid tumors (including breast cancer).

#### 4.1.2. CTLA-4

CTLA-4, known as CD152, is a leukocyte differentiation antigen. Such a protein is a transmembrane receptor on T cells that shares the B7 ligand with CD28, which participate in the negative regulation of immune response [64]. Furthermore, CTLA-4 has been proven to act as an efficient therapeutic target in cancer treatment. For instance, ipilimumab was the first CTLA-4 inhibitor for treating metastatic melanoma approved by U.S. Food and Drug Administration (FDA) in 2011 [65]. Its application has now been expanded to renal cell carcinoma, hepatocellular carcinoma, and colorectal cancer [66,67,68]. Moreover, research has further shown that preoperative cryoablation and single-dose ipilimumab are safe for use in monotherapy or in combination while treating early-stage breast cancer [69]. Another CTLA-4 inhibitor, tremelimumab, has also displayed therapeutic potential for treating breast cancer according to recent clinical evidence [70]. An ongoing phase II study is also evaluating the combination therapy of nivolumab (PD-1 inhibitor) and ipilimumab (CTLA-4) in patients with breast cancer (NCT03789110).

#### 4.1.3. TIM-3

T cell immunoglobulin and mucin domain 3 (TIM-3) (or CD366), also named as hepatitis A virus cellular receptor 2 (HAVCR2), is also a transmembrane protein. Ligands of TIM3 include galactin-9 (Gal-9), carcinoembryonic antigen cell adhesion molecule-1 (CEACAM-1), high mobility group protein B1 (HMGB1), and phosphatidylserine (PS) [71]. The TIM-3/Gal-9 axis has been found to modulate tumor immunity by negatively regulating T cell immunity [71]. Evidence showed that the blockade of TIM-3 can inhibit the proliferation, migration, and invasion of breast cancer cells, indicating that TIM-3 is an effective therapeutic target [72]. Several TIM-3 inhibitors, as novel ICIs, including sabatolimab and cobolimab, have been evaluating by certain ongoing clinical trials [73,74]. Moreover, RAPA-201 therapy, which has been thought to block the expression of PD-1, CTLA-4, TIM-3, and LAG3, thereby improving the immune suppressive tumor microenvironment, is currently being trialed in a phase I/II study for patients with solid cancer (including breast cancer) (NCT05144698). However, the clinical application of TIM-3 inhibitor in breast cancer treatment still requires more supportive evidence.

#### 4.1.4. LAG-3

Lymphocyte-activation gene 3 (LAG-3) is a type I transmembrane protein which mainly expressed in activated T cells, NK cells, B cells, and plasma cell dendritic cells [75]. Such an immune checkpoint has been found to regulate the activity of T cells by binding to the major histocompatibility complex II (MHC-II) and other ligands [76]. Evidence showed that the low expression of LAG-3 was substantially linked to longer relapse-free and overall survival times, making relatlimab, which acts as a LAG3 inhibitor, a promising anti-cancer agent [77]. For details, in 2022, the U.S. FDA approved the combination therapy of relatlimab and nivolumab for adults and children 12 years of age and older with unresectable or metastatic melanoma, setting the milestone for the application of LAG-3 inhibitors, as a novel ICI, in cancer treatment [78]. Currently, more LAG-3 immune checkpoint inhibitors (including tebotelimab, relatlimab, IMP321, etc.) have been trialed and presented strong efficacy against solid tumors including melanoma, breast cancer, NSCLC, and gastric cancer [79,80,81,82,83]. Such an agent has also displayed a good anti-tumor synergistic effect for PD-1/PD-L1 inhibitor in clinical trials [79,80,81,82,83].

#### 4.1.5. BTLA

BTLA (B and T lymphocyte attenuator, CD272) belongs to the CD28 family and is an important immune checkpoint molecule which is expressed in activated T and B lymphocytes [84]. Herpesvirus entry mediator (HVEM), which is widely expressed in various cell types (including in breast cancer) and participates in immune homeostasis, is the ligand of BTLA [85,86]. Evidence has also indicated that targeting BTLA (or HVEM) is a very promising novel immunotherapy for breast cancer treatment [85,86]. As a potential therapeutic target for cancer immunotherapy, BTLA is similar to PD-1 and CTLA-4, but possesses different functions [87]. Icatolimab (TAB004/JS004), as the first BTLA monoclonal antibody, is currently being tested in different clinical trials for cancer treatment (such as studies NCT04278859, NCT04137900, and NCT04477772) [88]. So far, icatolimab monotherapy, or in combination with triprilimab, was well tolerated and showed initial clinical efficacy in the treatment of relapsed/refractory lymphoma, [89].

#### 4.1.6. IDO1, 2

Indoleamine-2,3-dioxygenase (IDO)1 and IDO2 are two catabolizing enzymes which are closely related to tryptophan; they are induced under inflammatory conditions and have been found to regulate immune responses. The IDO1 is widely expressed in both immune and non-immune tissues, while IDO2 is restrictively expressed in liver, kidney, and antigen-presenting cells (dendritic cells and B cells). They have been discovered to catalyze the metabolism of tryptophan through the canine urine pathway, along with the unrelated enzyme tryptophan dioxygenase (TDO) [90]. Moreover, IDO1 and IDO2 have been considered to possess different abilities in anti-cancer immunity regulation [91]. Additionally, IDO1 and IDO2 are upregulated in many tumors, including breast cancer, and the overexpression of IDO1 has been found to be associated with poor prognosis [92]. As reported by a phase I/II trial (NCT02178722), a novel combined therapy of epacadostat (IDO1 inhibitor) and pembrolizumab (PD-1 inhibitor) was well tolerated and showed antitumor activity in patients with triple-negative breast cancer [55]. Consequently, in 2017, IDO1 inhibitor was then praised as the next hot research spot for cancer immunotherapy by the American Society of Clinical Oncology [93]. Furthermore, recent evidence from a wet-lab has further shown that the IDO1/2 dual-target inhibitor 4t has a superior antitumor efficacy compared to epacadostat [94], indicating that such a therapeutic strategy is worthy of further research and development investment.

#### 4.1.7. TIGIT

TIGIT (T cell immunoglobulin and ITIM domain), also named as WUCAM, Vstm3, and VSIG9, has been reported to possess the ability of suppressing both T cell-driven inflammation and T cell and NK cell-dependent anti-tumor immunity, as a coinhibitory receptor and a novel immune checkpoint [95,96]. Its expression can be upregulated by activated T cells, natural killer cells, and regulatory T cells. In addition, TIGIT has been found to bind three ligands, namely CD155 (PVR), CD112 (PVRL2, nectin-2), and CD113 (PVRL3), which are expressed by tumor cells, antigen-presenting cells, and also T-lymphocytes in the tumor microenvironment [95,96,97]. Furthermore, TIGIT has been found to affect innate and adaptive immunity through a variety of mechanisms. For instance, the binding of TIGIT and CD155 can cause the CD155-expressing dendritic cells to be tolerogenic and reduce the production of cytokines IL-12 and IL-10; additionally, TIGIT can also inhibit the degranulation of NK cells, the production of cytokine, and NK cell-mediated tumor cytotoxicity [98]. Based on those findings, TIGIT monoclonal antibodies (such as tiragolumab, vibostolimab, ociperlimab) and multiple immune checkpoint blockade therapies (including PD-1/TIGIT, and PD-1/TIGIT/LAG-3), have all been developed for cancer treatment, and some are currently in ongoing clinical trials [56,57,99,100,101,102]. In particular, high-frequency PD-1−TIGIT + CD8+ tumor-infiltrating lymphocytes (TILs) were observed in TNBC cells, making the strategy of dual PD-1/TIGIT blockade a promising therapeutic approach via increasing CD8+ TIL function in TNBC, which may further improve the immunotherapy for breast cancer [103].

#### 4.1.8. PD-L2

Programmed cell death ligand-2 (PD-L2) (known as CD273), is a member of the B7 family, and is another important ligand of PD-1 and, therefore, to affect the function of immune cells [104]. Furthermore, PD-L2 can competitively bind to PD-1 with a stronger affinity than PD-L1 to PD-1, thereby inhibiting the immune activation of T cells through downstream signal pathways which are related to tyrosine phosphatase 2 (SHP-2), lymphocyte-specific protein tyrosine kinase (LcK), phosphatidylinositol 3 (PI3K), and mitogen-activated protein kinase (MAPK) [105]. Additionally, PD-L2 has also been considered as a promising biomarker for risk assessment in patients with TNBC [105]. Moreover, agents which can selectively block dual targets, namely PD-L1 and PD-L2, have been intended to be developed for the treatment of a variety of advanced solid tumors, based on the fact that inhibition of PD-L2 can restore the ability of the immune system to recognize and kill cancer cells [106,107]. For instance, the combination therapy of DKY709 and PDR001 (a humanized IgG4 monoclonal antibody directed against PD-1 that blocks the binding of PD-L1 and PD-L2) is being tested in patients with triple-negative breast cancer in an ongoing phase I trial (NCT03891953). The combination therapy of ATRC-101-A01 and pembrolizumab is also being trialed in patients with breast cancer in a phase I study (NCT04244552).

#### 4.1.9. B7-H6

Natural killer cell cytotoxicity receptor 3 ligand 1 (NCR3LG1, also known as B7-H6) is a member of the B7 family which can regulate the T cell-mediated immune response [58]. It has been reported that, in TNBC cells (MDA-MB-231), B7-H6 knockdown can induce apoptosis and inhibit the proliferation of cancer cells, leading to the suppression of tumor development by regulating the T cell-mediated immune response, which indicated that B7-H6 possesses the potential to play a significant role in the treatment of TNBC [108].

### 4.2. Selected MicroRNA Targeted Multiple Immune Checkpoints in Breast Cancer

Based on the existing evidence, miRNAs which may regulate more than one immune checkpoint in breast cancer cells have been selected for potential immunotherapy application, as shown below (Table 1 and Figure 2). 

#### 4.2.1. MiR-93-5p

MiR-93-5p is a small noncoding RNA which acts as a gene regulator in numerous cancers [26]. It has been reported that overexpressed miR-93-5p can reduce the expression of PD-L1 and suppress the growth of MDA-MB-231 cells [26]. Moreover, miR-93-5p has also been shown to potentially regulate the immune evasion of breast cancer cells by targeting PD-L1, PD-L2, and B7-H6, based on a data analysis [25]. In colorectal cancer cells, miR-93-5p has been proven to downregulate the expression of PD-L1 by binding to the 3′-UTR region of PD-L1, and also to decrease the migration, invasion, and the immune evasion of cancer cells [24]. Therefore, miR-93-5p, as a circulating miRNA with multiple targets of immune checkpoints (including PD-L1, PD-L2, and B7-H6), has been considered to possess the potential in breast cancer immunotherapy, by a possible mechanism of reducing the expression of PD-L1, PD-L2, and B7-H6. However, additional research is still required for its application in cancer immunotherapy (Figure 2).

#### 4.2.2. MiR-149-3p

As a cancer suppressor gene the overexpression of miR-149-3p, has also been reported to decrease proliferation, migration, and invasion, and to trigger apoptosis in cells in different cancers (including breast cancer) [109]. Moreover, evidence has displayed that miR-149-3p can regulate the expression of PD-1, TIM-3, and BTLA by binding to the 3′UTRs of their mRNAs, thereby preventing tumor immune escape [27]. In addition, overexpressed miR-149-3p has been found to decrease T cell apoptosis and to weaken the miRNA markers of T cell exhaustion, which increased CD8+ T cells’ ability to destroy 4T1 breast cancer cells [27]. Thus, miR-149-3p may also be applied in breast cancer immunotherapy via regulating the expression of PD-1, TIM-3, and BTLA (Figure 2).

#### 4.2.3. MiR-195/MiR-497

As reported in previous studies, the expression of miR-195/miR-497 differs in normal and breast cancer tissues, making miR-195/miR-497 a potential biomarker for breast cancer treatment [110,111]. Further bioinformatic analysis indicated that miR-195/miR-497 can potentially regulate the immune evasion of breast cancer cells by targeting PD-L1 and B7-H6 [25]. For instance, miR-195/miR-497 mimicking transfection has been found to dramatically decrease the mRNA (messenger RNA) and protein expression of PD-L1 in MDA-MB-231 cells [25]. Additionally, the transfection of miR-195 has also been reported to downregulate the mRNA and protein expression of PD-L1, therefore reducing the proliferation and migration of OCI-Ly10 cells (diffuse large B cell lymphoma), and also preventing OCI-Ly10 cells from immune escape while raising apoptotic ratio in those cells [28]. Additionally, it has also been demonstrated that miR-195/miR-497 may target PD-L1 and B7-H6 after analyzing the transcriptome profiling (microRNA expression quantification) data from the TCGA database [25]. Thus, miR-195/miR-497 has been recommended as a potential candidate for multiple immune checkpoint blockade therapy in breast cancer by regulating the expression of PD-L1 and B7-H6. However, more research investment is still required for revealing its potential mechanisms and functions (Figure 2).

#### 4.2.4. MiR-5119

MiR-5119 has been reported to act as a regulator for anti-tumor immunity in breast cancer [29]. It has been found that engineering DCs (dendritic cells) and miR-5119, when employed as a novel combined therapeutic strategy, can simultaneously suppress multiple negative regulatory molecules, including IR (inhibitory receptor) ligands, such as PD-L1 and IDO2, in DCs, thereby improving anti-tumor immune response, upregulating cytokine production, and reducing the apoptosis of T cells in breast cancer cells [29]. Furthermore, MiR-5119 can also act as a potential regulator of PD-L1 in immune tolerance of tissue transplants [112]. Briefly, miR-5119 may target specific sequences in the 3′ UTRs of PD-L1 and IDO2 genes, therefore, suppressing the synthesis of both proteins [29]. As shown in a dual-luciferase reporter assay, miR-5119 mimic can reduce luciferase activity compared to the control miRNA, indicating that miR-5119 can directly bind and influence the expression of the PD-L1 gene [29]. Hence, miR-5119 has also been considered as a promising sensitizer for ICIs-based immunotherapy, even though further research work should be performed before its clinical application.

#### 4.2.5. MiR-138-5p

As a tumor suppressor miRNA, miR-138-5p has already been reported to suppress the metastasis, proliferation, and epithelial–mesenchymal transmission in different types of cancer cells [113,114]. Recently, many researches have also indicated that miR-138-5p can regulate the expression of certain immune checkpoints [115]. For instance, in breast cancer MDA-MB-231 cells, it has been reported that PD-L1 expression can be reduced by miRNA-138-5p transfection, and then the suppressed PD-L1 expression can further restrain the exhaustion of T cells, which inhibits proliferation, cloning, and migration of cancer cells [31]. Evidence has also found that miR-138-5p can exert multiple anti-tumor effects and immunostimulatory effects by targeting PD-1 and CTLA-4 in oral squamous cell carcinoma [30]. However, a similar function of miR-138-5p has not yet been confirmed in breast cancer.

#### 4.2.6. MiR-100-5p

MiR-100-5p is a member of the miR-100 family and is aberrantly expressed in different cancers [116,117]. Similar to miR-138-5p, miR-100-5p has also been proven to be a tumor suppressor by regulating cell death, the cell cycle, cell differentiation, proliferation, invasion, and migration of cancer cells [116]. For breast cancer, it has also been found that miR-100-5p transfection can promote cell death while impairing cell proliferation, migration, and invasion [116]. Furthermore, recent evidence has preliminarily revealed that the targets of miR-100-5p are possibly enriched in PD-1 and PD-L1 pathways, indicating the potential regulatory functions of miR-100-5p on PD-1 and PD-L1 [33]. Moreover, the expression of miR-100-5p has been reported to be negatively correlated with the expressions of PD-L1 and PD-L2 in bladder cancer, indicating a potential regulatory loop involving miR-100-5p, PD-L1, and PD-L2 [32]. Hence, miR-100-5p may also possess the potential to regulate the expression of PD-1, PD-L1, and PD-L2 in breast cancer.

#### 4.2.7. MiR-200a

MiR-200a is a member of miR-200 family and is abnormally overexpressed in breast cancer tissues [33]. Stomach cancer patients who have a higher expression level of miR-200a have a better prognosis compared to those who have lower expression level of miR-200a, indicating its important role in prognostic prediction [34]. Furthermore, miR-200a has also been found to be negatively associated with CD86 in gastrointestinal cancer as a potential therapeutic targets [34]. As for breast cancer, the excellent diagnostic and therapeutic performance held by the microRNA has also been reported [118,119]. Moreover, as for breast cancer, recent evidence has preliminarily displayed that the genes potentially targeted by miR-200a are enriched in the PD-1 and PD-L1 pathways, indicating the potential ability of miR-200a to regulate those immune checkpoints [33].

#### 4.2.8. MiR-21-5p

As the most common and strongly upregulated miRNA in glioblastoma, miR-21-5p has also been found to be abnormally overexpressed in different cancers including breast cancer [120]. Evidence has showed that suppression of miR-21-5p expression can reduce the migration, proliferation, and invasion of breast cancer cells [121]. In addition, potential pathways of miR-21-5p were predicted to mainly focus on PD-L1 and PD-1 immune checkpoints in breast cancer [33]. Moreover, transfection of miR-21-5p has also been reported to significantly elevate the expression levels of CTLA-4 and LAG3 miRNA in head and neck squamous cell carcinoma [35]. Therefore, the potential ability of miR-21-5p to target multiple immune checkpoints has been preliminarily displayed, and awaits further exploration.

#### 4.2.9. MiR-4443

In lung cancer patients, miR-4443 has been shown to potentially affect the expression of CTLA-4 and TIGIT, and the overall survival of patients by regulating the ZC3H12D-hsa-miR-4443-ENST00000630242 axis [36]. Therefore, miR-4443 was chosen as a potential candidate. However, its effects on breast cancer require further investigation.

### 4.3. Oncogenice or Tumor Suppressor Roles of Selected MicroRNAs

The miRNAs involved in cancer have been classified as two groups, one of which is the onco-miRNAs (oncoMiRs) which have been found to induce tumorigenesis and tumor progression, and these genes are abnormally and highly expressed in cancer cells. The other group is onco-suppressor miRNAs. This group of genes is frequently downregulated in cancer cells and can inhibit the cancer-related phenotype of cells [122]. Therefore, immune checkpoint targeting miRNAs which have already been classified may provide a more specific and clearer perspective in their application in cancer treatment.

For instance, overexpression of miR-93-5p has been found to diminish PD-L1 expression and inhibit MDA-MB-231 cell proliferation [26]. Furthermore, miR-93-5p can also control breast cancer cell migration by inhibiting the epithelial–mesenchymal transition (EMT) of the cells [123]. Thus, miR-93-5p is a promising onco-suppressor miRNA for the treatment of breast cancer.

Evidence also reported that overexpression of miR-149-3p can reduce proliferation, migration, and invasion, and can induce death in cancer cells (including breast cancer), indicating the potential of miR-149-3p as an onco-suppressor miRNA in breast cancer [109].

As for miR-100-5p, overexpression of such an onco-suppressor miRNA has been found to promote cell death and reduce proliferation, migration, and invasion of breast cancer cells [116]. Furthermore, it has been proved that miR-100-5p can suppress the migration and invasion of MDA-MB-231 breast cancer cells by targeting FZD-8 and inhibiting the Wnt/beta-catenin pathway [124]. As we presented above, miR-100-5p possesses the potential to regulate the expression of PD-1 and PD-L1, thus, exerting anti-tumor effects [33]. Therefore, miR-100-5p may hold multiple anti-cancer effects in breast cancer treatment.

In addition, the miR-200 family has been reported to regulate the transition between breast cancer-stem-cell-like and non-stem-cell-like phenotypes, and also modulate cellular plasticity and tumor suppression in breast cancer [125]. Furthermore, evidence has also shown that the expression of the miR-200b/200a/429 cluster can significantly inhibit the proliferation in the murine claudin-low mammary tumor cell line [126]. Furthermore, in breast cancer cells, miR-200a/b has been proven to inhibit the epithelial–mesenchymal transition and tumor metastasis via association with p73 as a tumor suppressor [127]. More specifically, evidence has shown that miR-200a can exert the anti-cancer effect on triple-negative breast cancer cells by direct repression of the EPHA2 oncogene [128]. As mentioned above, miR-200a can also potentially regulate the expression of PD-1 and PD-L1 and, therefore, enhances the therapeutic effect of an immune checkpoint blockade [33]. Thus, miR-200a may exert multiple anti-cancer effects as a potential sensitizer for ICIs in breast cancer treatment.

Evidence also found that knockdown of miR-21-5p can reduce the cell growth and increase the apoptosis level of MCF-7 cells [121,129]. However, the potential pathways of miR-21-5p have been predicted to mainly focus on PD-L1 and PD-1 immune checkpoints [121]. Therefore, miR-21, as a oncoMiR, may also become an effective anti-tumor immunity regulator with extra effects in regulating cancer development.

As for the other IC-targeting miRNAs mentioned above, such as miR-195/miR-497, miR-5119, miR-138-5p, and miR-4443, their roles of being oncoMiRs or onco-suppressor miRNAs in breast cancer are still not yet confirmed [25,29,31,36]. Therefore, more research exploration is recommended in this field.

### 4.4. Side Effect and Solutions

One of the main limitations of the ICI therapy is the appearance of immune-related adverse events (irAEs), which often leads to the interruption of the treatment. These irAEs are related side effects caused by the intervention of checkpoint inhibitors which can stimulate the immune system to raise up a series of intense immunoreactions not just against cancer cells, but also self normal tissues. For example, PD-1 inhibitors can cause pneumonitis, arthralgia, and hypothyroidism. Colitis, hypophysitis and rash were common in patients treated with CTLA4 inhibitors [130]. The incidence of irAEs can further increase if two immune checkpoint inhibitors are combined [18,131]. For instance, evidence has shown that CTLA4 inhibitors combined with PD-1 or PD-L1 agents can bring about higher incidence and severity of irAEs, such as the combination of ipilimumab and nivolumab, irrespective of the primary cancer treated [132,133,134]. Furthermore, immune-related adverse events could also be triggered by miRNA-based therapeutics [135,136,137]. That was the main reason why only 10 obtainable miRNA drugs have been in clinical trials with none undergoing phase III, while over 60 siRNA drugs are in complete clinical trial progression, including two approvals (patisiran and givosiran) [135,136,137]. For instance, MRX34 (a miR-34a mimic) in a phase I clinical trial led to tested objects undergoing five serious immune-related adverse events, therefore, terminating the project [138]. Additionally, RG-101, an anti-miR-122 drug, succeeded in phase I but was discontinued in phase II due to the occurrence of a few cases of hyperbilirubinemia [139].

The phenomenon of “too many targets for miRNA effect” (TMTME) has been considered to answer the main adverse events from the miRNA therapeutics [140]. Evidence has shown that one single miRNA can targets hundreds of genes [140]. Therefore, TMTME has been considered as an inevitable consequence caused by miRNA treatment. In other words, unlike most approved drugs with limited targets (including ICIs and siRNA drugs), a miRNA has far more target sequences to bind and cause incomplete complementation (including circRNAs, long non-coding RNAs, protein-coding genes, etc.) [140]. Hence, numerous chain reactions which might further lead to physiological dysfunction or other diseases (including irAEs) could happen after the introduction or removal of miRNA in humans [135]. Some of those changes are unknown and unpredictable, which brings huge obstacle to miRNA multiple target treatment [135]. Furthermore, as shown in our analyses performed using the TargetScan platform, the total amount of predicted targets of selected miRNAs ranges from 59 to 7852, which theoretically can trigger massive chain reactions by regulating multiple signal pathways, therefore, making TMTME an inevitable issue.

Of course, efforts have been made to eliminate or weaken the existing obstacles that were caused by TMTME. One of the potential solutions is the application of a safe and targeted drug delivery system which is a pattern of specifically designed carriers, largely based on nanomedicine [141,142]. Such a nanoparticle system has been considered to weaken the obstacles caused by TMTME by decreasing systemic drug concentration and effective doses [135]. Those nanoparticles were built to encapsulate and deliver miRNAs to specific lesion sites with enhanced solubility and efficacy of drugs, and reduced interaction with untargeted tissues [141,142,143]. Thus, potentially, the number of unnecessary targets that are supposed to be affected by delivered miRNAs could also be reduced [141,142,143]. Certain miRNA delivery systems were also developed for breast cancer treatment. For instance, in triple-negative breast cancer treatment, hyaluronic acid–chitosan was built to deliver miR-34a mimics [144], and RNA-NPs (nanoparticles) decorated with EGFR-targeting aptamer, which was used to carry a miR-21 inhibitor [145]. A novel fabricated nano-complex, namely gold–nanoparticles (AuNPs), was developed for miR-206 for the treatment of breast cancer [146]. However, as a potential approach for dealing with the TMTME issue, more research and still required to further evaluate the relevant effects of nanoparticle systems. The application of a delivery system also has disadvantages. For example, such an approach will undoubtedly increase the cost, and the storage could become more difficult due to the crystallization process, which might cause drug expulsion from the nanoparticles [147,148,149,150]. Another promising technique is the embodiment of miRNA therapeutics into a biodegradable 3D matrix [151]. Such an approach can reduce the side effects by directly implantation into a local lesion as part of a surgical intervention. Briefly, it has been proved that an implanted 3D matrix can locally trigger a continuous, tissue-related, and comparatively moderate release of miRNA-based curatives [152,153]. In addition, ex vivo miRNA-based therapeutics have also been regarded as a potential option which may provide more immediate clinical effect without going through complicated metabolic processes [151,152]. Such a technique has been shown to improve the effectiveness of adoptive immune cell transfer approaches, including potentiating adoptive T cell treatments, by regulating genes involved in T cell activation and fitness [151,152]. Moreover, based on the potential curative effect possessed by certain miRNAs originated from medically relevant plant sources, a novel concept of oral administration of miRNA-based therapeutics that stem from plants has been provided as a potential solution [154]. However, many problems, such as the bioavailability of miRNAs contained in plant food and the regulatory capacity of plant miRNAs in mammalian cells, are still unclear [155,156]. Therefore, massive investigations are required before any related therapeutic application can be implemented. Approaches, such as topical, periocular, and systemic corticosteroids administration have also been developing for controlling related side effects [157,158,159]. Indeed, the selection of qualified and proper miRNA is most crucial. However, to achieve this goal, significant research is required, since it is a novel and innovative field.

Another potential solution is to introduce the application of miRNAs that can predict the response to an immune checkpoint blockade [160]. Recent evidence has found that some miRNAs, as complex regulators of gene expression which reflect immune status and activity, can be used to predict the potential clinical benefit of ICI therapy (shown in Appendix A) [160]. For instance, in patients with non-small cell lung cancer (NSCLC), certain miRNAs (miR-215-5p, miR-411-3p, miR-493-5p, miR-494-3p, miR-495-3p, miR-548j-5p, and miR-93-3p) have been found to be expressed differently between the “good responders” (overall survival, OS > 6 months) and the “poor responders” (OS < 6 months), and showed a promising predictive value for ICIs therapy [161]. Furthermore, in esophageal squamous cell carcinoma patients, after nivolumab treatment, low expression levels of miR-6885-5p, miR-4698, and miR-128-2-5p assisted in separating responders from non-responders [162,163]. It has been concluded that targeting and regulating the immune checkpoint proteins is a potential mechanism for miRNAs to predict the response to ICIs and the therapeutic effect. Furthermore, readily accessible pretreatment blood miRNAs may provide a more convenient way to perform the prediction, making miRNA-based prediction very crucial for cancer immunotherapy [160]. However, to the best of our knowledge, miRNAs for predicting ICIs therapy in breast cancer have not yet been reported. Therefore, more research investment is strongly required in this field [164].

Additionally, alternatives to miRNAs, as sensitizers for ICI-based treatment, also require investment in research. Angiogenesis blockade, for example, has been found to exert a better therapeutic effect when applied with an immune checkpoint inhibitor in solid cancers [165]. Evidence also reported that an appropriate angiogenesis inhibitor can alleviate immunosuppression and enhance immunity by pruning blood vessels that are pivotal for tumor progression, and by blocking negative immune signals via decreasing the level at immune checkpoints, thereby increasing the anti-/pro-tumor immune subset ratio and alleviating hypoxia by normalizing tumor vasculature. Furthermore, the efficacy of anti-angiogenic therapies can also be improved by ICIs via recruiting immune cell subtypes with an angio-modulatory function [165]. Moreover, recent evidence has reported that histone deacetylases 2 (HDAC2) blockade can regulate the level of T lymphocyte subsets, inhibit tumor immune suppression and, consequently, enhance the efficacy of PD-1/PD-L1 inhibitors by regulating the progress of membranal PD-L1 nuclear translocation, improving anti-cancer immunity [143,166,167]. Thus, HDAC2 inhibitors, as a novel type of anticancer drug, have also displayed their potential as a sensitizer for cancer immunotherapy [166]. Furthermore, CAXII (carbonic anhydrase XII) inhibitor is also considered as a potential sensitizer which has been reported to reduce the immunosuppressive stress mediated by hypoxic/acidic metabolism, to modulate the expression of CCL8, and to affect the functions of monocytes and macrophages, thereby improving anti-tumor immunity and enhancing the therapeutic effect of PD-1 inhibitors in solid cancer [168,169,170,171,172]. Other approaches from alternative medicine, such as human biofield therapy, have also been mentioned [173,174]. Certainly, the combination strategy of multiple ICI sensitizers may also provide a novel perspective for the future study of cancer treatments.

## 5. Conclusions

The application of ICI monotherapy and combined ICIs has shown promise in treating a variety of malignancies. However, many patients, including breast cancer patients, cannot obtain an ideal and long-lasting curative effect from such a therapeutic approach. Based on the regulating effect of miRNAs on the transcription of immune checkpoints, a strategy combining miRNA with ICIs has been considered as a very novel and exciting way to enhance the potency of immune checkpoint-specific monotherapy. The miRNAs targeting multiple immune checkpoint molecules are believed to enhance the efficacy of ICI by mimicking combination therapy. Nonetheless, risks, such as irAEs, may also come along with such promising but innovative therapeutic synergism. Therefore, this review not only selected potential microRNAs for further exploration, but also discussed the risks associated with applying such a therapeutic strategy and the potential solutions to those risks. This may realize a whole new possible form of cancer immunotherapy by applying miRNAs with multiple immune checkpoint molecules as a novel additive therapy for ICI.

## Figures and Tables

**Figure 1 cancers-15-00824-f001:**
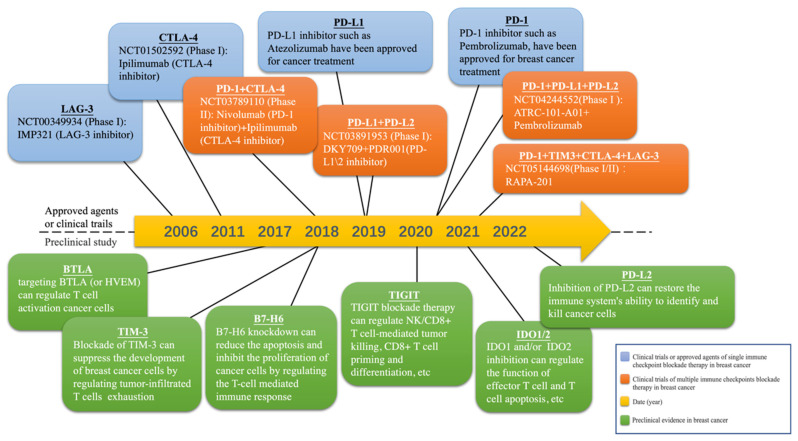
Milestones of selected immune checkpoint blockade therapy utilized in treating breast cancer. Currently, four clinical trials are testing different multiple immune checkpoint blockade therapy in breast cancer. The therapeutic value of seven novel immune checkpoint suppression methods in breast cancer treatment has been reported by preclinical studies.

**Figure 2 cancers-15-00824-f002:**
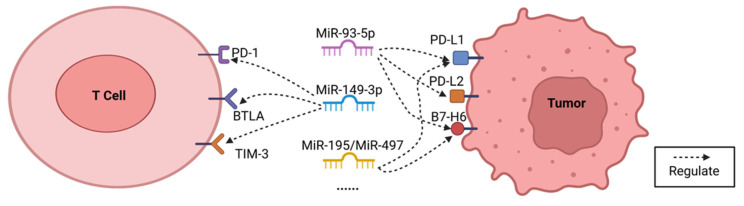
Selected miRNAs which can regulate multiple immune checkpoints, such as miR-93-5p, which can potentially affect the expression of PD-L1, PD-L2, and B7-H6; miR-149-3p, which can regulate the expression of PD-1, TIM-3, and BTLA. Furthermore, the expression of PD-L1 and B7-H6 can be modulated by miR-195/miR-497.

**Table 1 cancers-15-00824-t001:** Studies of selected miRNAs targeting multiple immune checkpoints and the total number of potential targets of selected miRNAs.

MiRNA	Targeted Immune Checkpoints	Tumor Type	Experimental Setting	Functional Mechanisms	References	Number of Predicted Targets	Conserved Sites and Poorly Conserved Sites
MiR-93-5p	B7-H6, PD-L1, PD-L2	Breast cancer, lung cancer, colorectal cancer	Database, in vitro, human sample	Reducing the expression of PD-1, PD-L1, PD-L2, and B7-H6	[24,25,26]	1385	2561
MiR-149-3p	PD-1, TIM-3, BTLA	Breast cancer	In vitro	Downregulating mRNAs for PD-1, TIM-3, and BTLA	[27]	7852	17186
MiR-195/MiR-497	PD-L1, B7-H6	Breast cancer, diffuse large B cell lymphoma	Database, in vitro	Reducing the expression of PD-L1, PD-L2, and B7-H6	[25,28]	1515	2456
MiR-5119	PD-L1, IDO2	Breast cancer	In vivo, in vitro	Downregulating the expression of PD-L1 and IDO2	[29]	3078	2537
MiR-138-5p	PD-L1, PD-1, CTLA-4	Breast cancer, oral squamous cell carcinoma	In vitro	Direct anti-tumoral effects and immunostimulatory effects by targeting PD-1 and CTLA-4	[30,31]	704	1093
MiR-100-5p	PD-L1, PD-1 PD-L2	Breast cancer, bladder cancer	Database, human sample	Downregulating the expression of PD-1, PD-L1, and PD-L2	[32,33]	59	62
MiR-200a	PD-L1, PD-1	Breast cancer, gastrointestinal cancer	Database, in vitro	Targeting PD-L1, PD-1, and CD86	[33,34]	905	1593
MiR-21-5p	PD-L1, PD-1, CTLA-4,LAG3	Breast cancer, head and neck squamous cell carcinoma	Database, in vitro	Upregulating PD-L1, PD-1, CTLA-4, and LAG3	[33,35]	384	552
MiR-4443	TIGIT, CTLA-4	Lung cancer	In vivo, In vitro, database	Targeting TIGIT and CTLA-4	[36,37]	4481	6052

**Table 2 cancers-15-00824-t002:** Application of selected immune checkpoint blockade therapy in cancer treatment.

	Ligand	Expression Location	Roles in Tumor Immunity	Potential Mechanisms	Approved Drugs or Candidates		Reference
					Targeting Receptors	Targeting Ligands	
PD-1	PD-L1/PD-L2	Lymphocytes including T, B, and NK/NKT cells	Suppressing T cell activation and proliferation in late phase; inducing T cell apoptosis.	Phosphorylated PD-1-ITIM/SHP2/SAP signaling; TCR signaling inhibition	Pembrolizumab (approved); pucotenlimab (approved); RAPA-201 (phase I/II); nivolumab+ ipilimumab (phase II); ATRC-101-A01 +pembrolizumab (phase I)	Atezolizumab (Approved); Durvalumab (Approved); ATRC-101-A01 + Pembrolizumab (Phase I); DKY709 + PDR001 (Phase I)	[40,41,42,43]
CTLA-4	CD80/CD86	Activated T cells	Inhibiting T cell activation in early phase	Phosphorylated CTLA4-YVKM/SHP2/RAS signaling; TGF-β/IDO inducing	Ipilimumab (approved); tremelimumab (approved); RAPA-201 (phase I/II); Nivolumab+ Ipilimumab (phase II)	NA	[44,45,46]
TIM-3	Galactin-9, CEACAM-1, HMGB1, PS	Tumor-infiltrating T cells, Tregs, DCs, monocytes, NK cells	Exhausting tumor-infiltrated T cells	Glycosylated TIM3/AKT/mTOR signaling; phosphorylated TIM3/NFAT/Bat signaling	Sabatolimab (phase III); MAS825 (phase III) RAPA-201 (phase I/II)	NA	[47,48]
LAG-3	MHC-II, galectin-3, LSECtin	Activated T cells, B, NK cells, DCs	Preventing CD4-MHC-II interaction; inhibiting CD4-dependent T cell function	Phosphorylated LAG-3-KIEELE/mediated reduction in calcium influx impairs TCR signaling	Relatlimab (approved); RAPA-201 (phase I/II)	NA	[46,49]
BTLA	HVEM	T, B, NK cells, macrophages, DCs	Holding back T cell over-activation	Phosphorylated BTLA-ITIM/ITSM/SHP2; inhibiting both TCR and CD28 signaling	Icatolimab (phase II)	NA	[50,51,52]
IDO1/2	AhR	Tumor cells, stromal cells, and immune cells in TME	Inhibiting the function of effector T cell and promoting Tregs; inducing T cell apoptosis	Catalyzing the oxidative cleavage of tryptophan; producing metabolite kynurenic acid	Epacadostat (phase II); epacadostat and pembrolizumab (phase I/II)	NA	[53,54,55]
TIGIT	CD155, CD112, CD113	T cells, Tregs, NKT cells	Inhibiting NK/CD8+ T cell-mediated tumor killing; affecting CD8+ T cell priming and differentiation; inducing immunosuppressive DCs	TIGIT/PVR/IL-10 and TGF-β signaling; TIGIT/CD155 ERK signaling	Tiragolumab (phase III); vibostolimab (phase III); ociperlimab (phase III)	NA	[56,57]
B7-H6	NKp30	Tumor cells	Regulating the T cell-mediated immune response	Helping NK cells to recognize abnormal cells	NA	NA	[58]

Abbreviation—NA, not available.

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
