# Peer review of "MicroRNAs with Multiple Targets of Immune Checkpoints, as a Potential Sensitizer for Immune Checkpoint Inhibitors in Breast Cancer Treatment"

_cancers, 2023, doi:10.3390/cancers15030824_

Round 1

Reviewer 1 Report

·      Authors should define ORR first time

·      In Methods, the word “the” at the beginning can be removed.

·      Are there any specific suggestions/speculations to drastically reduce the effect of TMTME or even suggestions for alternatives to microRNAs for cancer therapy.

Author Response

Response to Reviewer 1 Comments

Review 1

  1. Authors should define ORR first time.

Answer

Thank the reviewer so much for his/her careful comment. We have now added the explanation of objective response rate for ORR at its first appearance in manuscript, showed as followed: “In actuality, not only is the ORR (Objective response rate) induced by ICIs highly restricted but the number and severity of adverse effects caused by combinations of immune checkpoint inhibitors are frustrating” (Line 78, Page 2).

2.In Methods, the word “the” at the beginning can be removed.

Answer

Thank you for your thoughtful comment. We have deleted the word in Method section, showed as followed: “TargetScan database (https://www.targetscan.org/vert_80/)(TargetScan Release 8.0) was applied to predict the potential targets of selected miRNAs” (Line 101, Page 2).

  1. Are there any specific suggestions/speculations to drastically reduce the effect of TMTME or even suggestions for alternatives to microRNAs for cancer therapy.

Answer

Thank you so much for your excellent comments. To deal with the effect of TMTME (too many targets for miRNA effect), potential suggestions have already been described in the section of “Side-effect and solutions”. To further discuss such issues, we have now revised and updated whole new context in our manuscript (Line 479, Page 12- Line 532, Page 13), as showed below:

“Of course, for eliminating or weakening the existing obstacles that were caused by TMTME, efforts have been made to conquer such issue. One of the promising solutions is the application of safe and targeted drug delivery system which is a pattern of specifically designed carriers, largely based on nanomedicine[145, 146]. Such nanoparticle system has been considered to eliminate or weaken the obstacles caused by TMTME by recognizing the antigens and receptors on cancer cell membranes[139]. Those nanoparticles were built to encapsulate and deliver miRNAs to specific lesion sites with enhanced solubility and efficacy of drugs, and reduced interaction with untargeted tissues[145-147]. Certain miRNA delivery systems were also developed for breast cancer treatment. For instance, in triple-negative breast cancer treatment, hyaluronic acid‑chitosan was built to deliver miRNA‑34a mimics[148], and RNA‑NPs(nanoparticles) decorated with EGFR‑targeting aptamer was for carry miRNA‑21 inhibitor[149]. A novel fabricated nano-complex, gold-nanoparticles (AuNPs) was developed to miR-206 for treating breast cancer[150]. However, application of delivery system also has disadvantages. For example, such approach will undoubtedly increase the cost, and the storage could become more difficult due to the crystallization process which might cause drug expulsion from the nanoparticles[151, 152] [153, 154]. Besides that, another promising technique is the embodiment of miRNA therapeutics into a biodegradable 3D matrix[155]. Such approach can reduce the side-effect by directly implantation into local lesion as part of a surgical intervention. Briefly, it has been proved that implanted 3D matrix can locally trigger a continuous, tissue-related and comparatively moderate release of miRNA-based curatives[156, 157]. In addition, ex-vivo miRNA-based therapeutics has also been regarded as a potential option which may provide more immediate clinical effect without going through complicated metabolic processes[155, 156]. Such technic has been shown to improve the effectiveness of adoptive immune cell transfer approaches, including potentiating adoptive T cell treatments, by regulating genes involved in T cell activation and fitness[155, 156]. Moreover, based on the potential curative effect possessed by certain miRNAs originated from medically relevant plant sources, a novel concept of oral administration of miRNA-based therapeutics that stem from plants has been provided as a potential solution[158]. However, many problems such as the bioavailability of miRNAs contained in plant food, and the regulatory capacity of plant miRNAs in mammalian cells, are still unclear[159, 160]. Therefore, massive investigations are required before any related therapeutic application can be implemented. Approaches such as topical, periocular and systemic corticosteroids administration, have also been developing for controlling related side effects[161-163]. Indeed, the selection of qualified and proper miRNA is most crucial. However, to achieve this goal, abundant of research are required, since it is a novel and innovative field.

Another potential solution is to introduce the application of miRNAs that can predict the response to immune checkpoint blockade[164]. Recent evidence has found that, some miRNAs, as complex regulator of gene expression, which reflects immune status and activity, can be used to predict the potential clinical benefit of ICI therapy (shown in Supplementary file 1)[164]. For instance, in patients with non-small cell lung cancer (NSCLC), certain miRNAs (miR-215-5p, miR-411-3p, miR-493-5p, miR-494-3p, miR-495-3p, miR-548j-5p and miR-93-3p) have been found to express differently between the “good responders” (overall survival, OS>6 months) and the “poor responders” (OS < 6 months), and showed showing promising predictive function for ICIs therapy[165]. Furthermore, in esophageal squamous-cell carcinoma patients, after nivolumab treatment, low expression levels of miR-6885-5p, miR-4698, and miR-128-2-5passisted in separating responders from non-responders[166, 167]. It has been concluded that targeting and regulating the immune checkpoint protein is the potential mechanism for miRNAs to predict the response to ICIs and therapeutic effect. Besides that, readily accessible pretreatment blood miRNAs may provide a more convenient way perform the prediction, makes miRNAs based prediction a very crucial for cancer immunotherapy[164]. However, to our best knowledge, miRNAs for predicting ICIs therapy in breast cancer, have not been reported yet. Therefore, more research investment is strongly required in such field[168].”

As for suggestions for alternatives to microRNAs for cancer therapy, we also added corresponding new context and discussion as suggested (Line 533, Page 13 - Line 555, Page 14), as showed below:

Additionally, alternatives to microRNAs, as sensitizer for ICIs based treatment, also require investment in research. Angiogenesis blockade, for example, has been found to exert better therapeutic effect when applied with immune checkpoint inhibitor in solid cancers[169]. Evidence also reported that appropriate angiogenesis inhibitor can alleviate immunosuppression and enhance immunity by pruning blood vessels that are pivotal for tumor progression, and blocking negative immune signals via decreasing the level of immune checkpoints, thereby increasing the anti-/pro-tumor immune subset ratio and alleviating hypoxia by normalizing tumor vasculature. Besides that, the efficacy of anti-angiogenic therapies can also be improved by ICBs via recruiting immune cell subtypes with angio-modulatory function[169]. Moreover, recent evidence has reported that histone deacetylases 2 (HDAC2) blockade can regulate the level of T lymphocyte subsets, inhibit tumor immune suppression and consequently enhance the efficacy of PD-1/PD-L1 inhibitor by regulating the progress of membranal PD-L1nuclear translocation and improving anti-cancer immunity[147, 170, 171]. Thus, HDAC2 inhibitors, as a novel type of anticancer drug, has also displayed its potential as a sensitizer for cancer immunotherapy[170]. CAXII (Carbonic anhydrase XII) inhibitor is also considered as a potential sensitizer which has been reported to reduce the immunosuppressive stress mediated by hypoxic/acidic metabolism, modulate the expression of CCL8 and affect the functions of monocytes and macrophages, thereby improving anti-tumor immunity and enhancing the therapeutic effect of PD-1 inhibitors in solid cancer[172-176]. Other approaches from alternative medicine such as human biofield therapy, have also been mentioned[177, 178]. Certainly, the combination strategy of multiple ICIs sensitizers may also provide novel perspective for future study of cancer treatment.”

Furthermore, we have added fresh section named “Oncogenice or tumor suppressor roles of selected microRNAs” to specifically discussed the potential effects of those IC-targeting miRNAs which have already been classified as OncoMiRs or tumor suppressor miRNAs in breast cancer (from Line 400 in Page 11 to Line 443 in Page 11), as shown below:

Oncogenice or tumor suppressor roles of selected microRNAs

MiRNAs involved in cancer have been classified as two groups: one is the onco-miRNAs (oncoMiRs) which have been found to induce tumorigenesis and tumor progression. Those genes are abnormally and highly expressed in cancer cells; Another one is onco-suppressor miRNAs. This group of genes is frequently downregulated in cancer cells, and can inhibit the cancer-related phenotype of cells[123]. Therefore, immune checkpoint targeting miRNAs which have already been classified, may provide more specific and clear perspective in their application in cancer treatment.

For instance, overexpression of miR-93-5p has been found to diminish PD-L1 expression and inhibit MDA-MB-231 cell proliferation[27]. Also, miR-93-5p can also control breast cancer cell migration by inhibiting the Epithelial-mesenchymal transition (EMT) of the cells[124]. Thus, miR-93-5p is a promising onco-suppressor miRNAs in treatment of breast cancer.

Evidence also reported that overexpression of miR-149-3p can reduce proliferation, migration, invasion, and induce death in cancer cells (including breast cancer), indicating the potential of miR-149-3p as an onco-suppressor miRNAs in breast cancer[125].

As for miR-100-5p, overexpression of such onco-suppressor miRNA has been found to promote cell death and reduce proliferation, migration, and invasion of breast cancer cells[126]. Also, it has been proved that miR-100-5p can suppress the migration and invasion of MDA-MB-231 breast cancer cells by targeting FZD-8 and inhibiting Wnt/beta-catenin pathway[127]. As we presented above, miR-100-5p possesses the potential to regulate the expression of PD-1 and PD-L1 thus exerting anti-tumor effects[34]. Therefore, miR-100-5p may hold multiple anti-cancer effects in breast cancer treatment.

In addition, miR-200 family has been reported to regulate the transition between breast cancer-stem-cell-like and non-stem-cell-like phenotypes, and also modulate cellular plasticity and tumor suppression in breast cancer[128]. Furthermore, evidence has also displayed that the expression of miR-200b/200a/429 cluster can significantly inhibit the proliferation in murine claudin-low mammary tumor cell line[129]. Also, in breast cancer cells, miR-200a/b has been proved to inhibit the epithelial-mesenchymal transition and tumor metastasis via association with p73, as a tumor suppressor[130]. More specifically, evidence has showed that miR-200a can exert the anti-cancer effect on triple-negative breast cancer cells by direct repression of the EPHA2 oncogene[131]. As mentioned above, miR-200a can also potentially regulate the expression of PD-1 and PD-L1 therefore enhance the therapeutic effect of immune checkpoint blockade[34]. Thus, miR-200a may exert multiple anti-cancer effects as a potential sensitizer for ICIs in breast cancer treatment.

Evidence also found that knocked down of miR-21-5p can reduce the cell growth and increase the apoptosis level of MCF-7 cells[132, 133]. However, the potential pathways of miR-21-5p have been predicted to mainly focus on PD-L1 and PD-1 immune checkpoints[133]. Therefore, miR-21, as a oncoMiR, may also become an effective anti-tumor immunity regulator with extra effects in regulating cancer development.

As for other IC-targeting miRNAs mentioned above, such as miR-195/miR-497, miR-5119, miR-138-5p and miR-4443, their roles of being oncoMiRs or onco-suppressor miRNAs in breast cancer are still not been decided yet[26, 30, 32, 37]. Therefore, more research exploration is recommended in such field. ”

Moreover, The IC-targeting miRNAs summarized above, which have also been classified as oncoMiRs or onco-suppressor miRNAs, are listed below and shown in Supplementary file 1:

“miR-215-5p,miR-411-3p,miR-493-5p, miR-494-3p, miR-495-3p, miR-548j-5p and miR-93-3p: The expression of these seven miRNAs are different between responders with an overall survival(OS) > 6 months and the responders with an OS < 6 months in non-small cell lung cancer(NSCLC)  patients treated by nivolumab, showing good predicting for ICIs therapy[1].

miR-6885-5p, miR-4698, and miR-128-2-5p: In esophageal squamous-cell carcinoma patients treated by nivolumab , the three miRNAs can distinguish responders from non-responders[2].

miR-222:the melanoma patients advancing on ICI therapy have greater expression levels of miR-222 than the patients who have a clinical benefit from immune checkpoint blockage[3].

miR-320d, miR-320c, miR-320b: In NSCLC patients, baseline levels of the three miRNAs are significantly lower in responders to ICI therapy compared to non-responders[4].

miR-208a-5p,miR-574-5p: NSCLC patients with expression levels of miR-208a-5p and miR-574-5p below the median have a significantly longer OS following the start of immune-checkpoint blocking than patients with expression levels above the median[5].

let-7,miR-125a,miR-99b,miR-146b,miR-125b: Melanoma patients with low expression level of the five miRNAs undergoing immune-checkpoint blockade with either ipilimumab or nivolumab displayed a statistically significantly better OS compared to high expression level patients[6].

Reference:

  1. Halvorsen, A.R.; Sandhu, V.; Sprauten, M.; Flote, V.G.; Kure, E.H.; Brustugun, O.T.; Helland, A. Circulating microRNAs associated with prolonged overall survival in lung cancer patients treated with nivolumab. Acta Oncologica 2018, 57, 1225-1231, doi:10.1080/0284186x.2018.1465585.
  2. Sudo, K.; Kato, K.; Matsuzaki, J.; Takizawa, S.; Aoki, Y.; Shoji, H.; Iwasa, S.; Honma, Y.; Takashima, A.; Sakamoto, H.; et al. Identification of serum microRNAs predicting the response of esophageal squamous-cell carcinoma to nivolumab. Japanese Journal of Clinical Oncology 2020, 50, 114-121, doi:10.1093/jjco/hyz146.
  3. Galore-Haskel, G.; Nemlich, Y.; Greenberg, E.; Ashkenazi, S.; Hakim, M.; Itzhaki, O.; Shoshani, N.; Shapira-Fromer, R.; Ben-Ami, E.; Ofek, E.; et al. A novel immune resistance mechanism of melanoma cells controlled by the ADAR1 enzyme. Oncotarget 2015, 6, 28999-29015, doi:10.18632/oncotarget.4905.
  4. Peng, X.X.; Yu, R.Y.; Wu, X.; Wu, S.Y.; Pi, C.; Chen, Z.H.; Zhang, X.C.; Gao, C.Y.; Shao, Y.; Liu, L.; et al. Correlation of plasma exosomal microRNAs with the efficacy of immunotherapy in EGFR/ALK wild-type advanced non-small cell lung cancer. Journal for Immunotherapy of Cancer 2020, 8, doi:10.1136/jitc-2019-000376.
  5. Genova, C.; Coco, S.; Rossi, G.; Longo, L.; Chiorino, G.; Ostano, P.; Guana, F.; Metro, G.; Baglivo, S.; Ludovini, V.; et al. An exosomal miRNA signature as predictor of benefit from immune checkpoint inhibitors in non-small cell lung cancer. Annals of Oncology 2020, 31, S825-S826, doi:10.1016/j.annonc.2020.08.1591.
  6. Huber, V.; Vallacchi, V.; Fleming, V.; Hu, X.Y.; Cova, A.; Dugo, M.; Shahaj, E.; Sulsenti, R.; Vergani, E.; Filipazzi, P.; et al. Tumor-derived microRNAs induce myeloid suppressor cells and predict immunotherapy resistance in melanoma. Journal of Clinical Investigation 2018, 128, 5505-5516, doi:10.1172/jci98060."

Reviewer 2 Report

The manuscript review the MiRNAs and their roles in targeting immune checkpoint proteins.

The concept is very interesting. I have the following suggestion.

1) As previously published (PMID: 32342824), microRNAs are classified in breast cancer. Briefly explain (in text or in the form of a Table) the classification of miRNAs in breast cancer (OncoMiRs and tumor suppressor miRNAs). And mention if any IC-targeting miRNAs are in the previous list or if they are new. This will give the bigger picture.

2) It is interesting the newly approved Cadonilimab is explained in the manuscript. However,  the TNBC indication of atezolizumab was withdrawn in August 2021 ( PMID: 29438695). Please, highlight if miRNAs can be crucial in predicting the response to immune checkpoint blockades. If they can, please provide a list of such miRNAs.

Thank you

Author Response

Response to Reviewer 2 Comments

Review 2

The manuscript review the MiRNAs and their roles in targeting immune checkpoint proteins. The concept is very interesting. I have the following suggestion.

1) As previously published (PMID: 32342824), microRNAs are classified in breast cancer. Briefly explain (in text or in the form of a Table) the classification of miRNAs in breast cancer (OncoMiRs and tumor suppressor miRNAs). And mention if any IC-targeting miRNAs are in the previous list or if they are new. This will give the bigger picture.

Answer

Thank the reviewer so much for his/her excellent comment. We appreciate and totally agree the point of summarizing IC-targeting miRNAs which have already been classified as OncoMiRs or tumor suppressor miRNAs in breast cancer. Therefore, we have now added a section named “Oncogenice or tumor suppressor roles of selected microRNAs” to specifically discussed such issue (from Line 400 in Page 11 to Line 443 in Page 11) (showed as bellow):

Oncogenice or tumor suppressor roles of selected microRNAs

MiRNAs involved in cancer have been classified as two groups: one is the onco-miRNAs (oncoMiRs) which have been found to induce tumorigenesis and tumor progression. Those genes are abnormally and highly expressed in cancer cells; Another one is onco-suppressor miRNAs. This group of genes is frequently downregulated in cancer cells, and can inhibit the cancer-related phenotype of cells[123]. Therefore, immune checkpoint targeting miRNAs which have already been classified, may provide more specific and clear perspective in their application in cancer treatment.

For instance, overexpression of miR-93-5p has been found to diminish PD-L1 expression and inhibit MDA-MB-231 cell proliferation[27]. Also, miR-93-5p can also control breast cancer cell migration by inhibiting the Epithelial-mesenchymal transition (EMT) of the cells[124]. Thus, miR-93-5p is a promising onco-suppressor miRNAs in treatment of breast cancer.

Evidence also reported that overexpression of miR-149-3p can reduce proliferation, migration, invasion, and induce death in cancer cells (including breast cancer), indicating the potential of miR-149-3p as an onco-suppressor miRNAs in breast cancer[125].

As for miR-100-5p, overexpression of such onco-suppressor miRNA has been found to promote cell death and reduce proliferation, migration, and invasion of breast cancer cells[126]. Also, it has been proved that miR-100-5p can suppress the migration and invasion of MDA-MB-231 breast cancer cells by targeting FZD-8 and inhibiting Wnt/beta-catenin pathway[127]. As we presented above, miR-100-5p possesses the potential to regulate the expression of PD-1 and PD-L1 thus exerting anti-tumor effects[34]. Therefore, miR-100-5p may hold multiple anti-cancer effects in breast cancer treatment.

In addition, miR-200 family has been reported to regulate the transition between breast cancer-stem-cell-like and non-stem-cell-like phenotypes, and also modulate cellular plasticity and tumor suppression in breast cancer[128]. Furthermore, evidence has also displayed that the expression of miR-200b/200a/429 cluster can significantly inhibit the proliferation in murine claudin-low mammary tumor cell line[129]. Also, in breast cancer cells, miR-200a/b has been proved to inhibit the epithelial-mesenchymal transition and tumor metastasis via association with p73, as a tumor suppressor[130]. More specifically, evidence has showed that miR-200a can exert the anti-cancer effect on triple-negative breast cancer cells by direct repression of the EPHA2 oncogene[131]. As mentioned above, miR-200a can also potentially regulate the expression of PD-1 and PD-L1 therefore enhance the therapeutic effect of immune checkpoint blockade[34]. Thus, miR-200a may exert multiple anti-cancer effects as a potential sensitizer for ICIs in breast cancer treatment.

Evidence also found that knocked down of miR-21-5p can reduce the cell growth and increase the apoptosis level of MCF-7 cells[132, 133]. However, the potential pathways of miR-21-5p have been predicted to mainly focus on PD-L1 and PD-1 immune checkpoints[133]. Therefore, miR-21, as a oncoMiR, may also become an effective anti-tumor immunity regulator with extra effects in regulating cancer development.

As for other IC-targeting miRNAs mentioned above, such as miR-195/miR-497, miR-5119, miR-138-5p and miR-4443, their roles of being oncoMiRs or onco-suppressor miRNAs in breast cancer are still not been decided yet[26, 30, 32, 37]. Therefore, more research exploration is recommended in such field. ”

Moreover, The IC-targeting miRNAs which have also been classified as oncoMiRs or onco-suppressor miRNAs, are listed below and shown in Supplementary file 1:

“miR-215-5p,miR-411-3p,miR-493-5p, miR-494-3p, miR-495-3p, miR-548j-5p and miR-93-3p: The expression of these seven miRNAs are different between responders with an overall survival(OS) > 6 months and the responders with an OS < 6 months in non-small cell lung cancer(NSCLC)  patients treated by nivolumab, showing good predicting for ICIs therapy[1].

miR-6885-5p, miR-4698, and miR-128-2-5p: In esophageal squamous-cell carcinoma patients treated by nivolumab , the three miRNAs can distinguish responders from non-responders[2].

miR-222:the melanoma patients advancing on ICI therapy have greater expression levels of miR-222 than the patients who have a clinical benefit from immune checkpoint blockage[3].

miR-320d, miR-320c, miR-320b: In NSCLC patients, baseline levels of the three miRNAs are significantly lower in responders to ICI therapy compared to non-responders[4].

miR-208a-5p,miR-574-5p: NSCLC patients with expression levels of miR-208a-5p and miR-574-5p below the median have a significantly longer OS following the start of immune-checkpoint blocking than patients with expression levels above the median[5].

let-7,miR-125a,miR-99b,miR-146b,miR-125b: Melanoma patients with low expression level of the five miRNAs undergoing immune-checkpoint blockade with either ipilimumab or nivolumab displayed a statistically significantly better OS compared to high expression level patients[6].

Reference:

  1. Halvorsen, A.R.; Sandhu, V.; Sprauten, M.; Flote, V.G.; Kure, E.H.; Brustugun, O.T.; Helland, A. Circulating microRNAs associated with prolonged overall survival in lung cancer patients treated with nivolumab. Acta Oncologica 2018, 57, 1225-1231, doi:10.1080/0284186x.2018.1465585.
  2. Sudo, K.; Kato, K.; Matsuzaki, J.; Takizawa, S.; Aoki, Y.; Shoji, H.; Iwasa, S.; Honma, Y.; Takashima, A.; Sakamoto, H.; et al. Identification of serum microRNAs predicting the response of esophageal squamous-cell carcinoma to nivolumab. Japanese Journal of Clinical Oncology 2020, 50, 114-121, doi:10.1093/jjco/hyz146.
  3. Galore-Haskel, G.; Nemlich, Y.; Greenberg, E.; Ashkenazi, S.; Hakim, M.; Itzhaki, O.; Shoshani, N.; Shapira-Fromer, R.; Ben-Ami, E.; Ofek, E.; et al. A novel immune resistance mechanism of melanoma cells controlled by the ADAR1 enzyme. Oncotarget 2015, 6, 28999-29015, doi:10.18632/oncotarget.4905.
  4. Peng, X.X.; Yu, R.Y.; Wu, X.; Wu, S.Y.; Pi, C.; Chen, Z.H.; Zhang, X.C.; Gao, C.Y.; Shao, Y.; Liu, L.; et al. Correlation of plasma exosomal microRNAs with the efficacy of immunotherapy in EGFR/ALK wild-type advanced non-small cell lung cancer. Journal for Immunotherapy of Cancer 2020, 8, doi:10.1136/jitc-2019-000376.
  5. Genova, C.; Coco, S.; Rossi, G.; Longo, L.; Chiorino, G.; Ostano, P.; Guana, F.; Metro, G.; Baglivo, S.; Ludovini, V.; et al. An exosomal miRNA signature as predictor of benefit from immune checkpoint inhibitors in non-small cell lung cancer. Annals of Oncology 2020, 31, S825-S826, doi:10.1016/j.annonc.2020.08.1591.
  6. Huber, V.; Vallacchi, V.; Fleming, V.; Hu, X.Y.; Cova, A.; Dugo, M.; Shahaj, E.; Sulsenti, R.; Vergani, E.; Filipazzi, P.; et al. Tumor-derived microRNAs induce myeloid suppressor cells and predict immunotherapy resistance in melanoma. Journal of Clinical Investigation 2018, 128, 5505-5516, doi:10.1172/jci98060."

2) It is interesting the newly approved Cadonilimab is explained in the manuscript. However,  the TNBC indication of atezolizumab was withdrawn in August 2021 ( PMID: 29438695). Please, highlight if miRNAs can be crucial in predicting the response to immune checkpoint blockades. If they can, please provide a list of such miRNAs.

Answer:

Thank you so much for the brilliant suggestion. We have specifically added content to discuss and emphasize the importance of exploring miRNAs that can predict the response to immune checkpoint blockades, since it was a whole new filed for breast cancer ICI therapy (from Line 515 in Page 13 to Line 532 in Page 13) (showed as bellow):

“Another potential solution is to introduce the application of miRNAs that can predict the response to immune checkpoint blockade[164]. Recent evidence has found that, some miRNAs, as complex regulator of gene expression, which reflects immune status and activity, can be used to predict the potential clinical benefit of ICI therapy (shown in Supplementary file 1)[164]. For instance, in patients with non-small cell lung cancer (NSCLC), certain miRNAs (miR-215-5p, miR-411-3p, miR-493-5p, miR-494-3p, miR-495-3p, miR-548j-5p and miR-93-3p) have been found to express differently between the “good responders” (overall survival, OS>6 months) and the “poor responders” (OS < 6 months), and showed showing promising predictive function for ICIs therapy[165]. Furthermore, in esophageal squamous-cell carcinoma patients, after nivolumab treatment, low expression levels of miR-6885-5p, miR-4698, and miR-128-2-5passisted in separating responders from non-responders[166, 167]. It has been concluded that targeting and regulating the immune checkpoint protein is the potential mechanism for miRNAs to predict the response to ICIs and therapeutic effect. Besides that, readily accessible pretreatment blood miRNAs may provide a more convenient way perform the prediction, makes miRNAs based prediction a very crucial for cancer immunotherapy[164]. However, to our best knowledge, miRNAs for predicting ICIs therapy in breast cancer, have not been reported yet. Therefore, more research investment is strongly required in such field[168].

Another potential solution is to introduce the application of miRNAs that can predict the response to immune checkpoint blockade[164]. Recent evidence has found that, some miRNAs, as complex regulator of gene expression, which reflects immune status and activity, can be used to predict the potential clinical benefit of ICI therapy (shown in Supplementary file 1)[164]. For instance, in patients with non-small cell lung cancer (NSCLC), certain miRNAs (miR-215-5p, miR-411-3p, miR-493-5p, miR-494-3p, miR-495-3p, miR-548j-5p and miR-93-3p) have been found to express differently between the “good responders” (overall survival, OS>6 months) and the “poor responders” (OS < 6 months), and showed showing promising predictive function for ICIs therapy[165]. Furthermore, in esophageal squamous-cell carcinoma patients, after nivolumab treatment, low expression levels of miR-6885-5p, miR-4698, and miR-128-2-5p assisted in separating responders from non-responders[166, 167]. It has been concluded that targeting and regulating the immune checkpoint protein is the potential mechanism for miRNAs to predict the response to ICIs and therapeutic effect. Besides that, readily accessible pretreatment blood miRNAs may provide a more convenient way perform the prediction, makes miRNAs based prediction a very crucial for cancer immunotherapy[164]. However, to our best knowledge, miRNAs for predicting ICIs therapy in breast cancer, have not been reported yet. Therefore, more research investment is strongly required in such field[168].”

Furthermore, to further discuss the potential approaches to deal with effect of TMTME (too many targets for miRNA effect), we have now revised and updated whole new context in our manuscript (Line 479, Page 12- Line 514, Page 13), as showed below:

“Of course, for eliminating or weakening the existing obstacles that were caused by TMTME, efforts have been made to conquer such issue. One of the promising solutions is the application of safe and targeted drug delivery system which is a pattern of specifically designed carriers, largely based on nanomedicine[145, 146]. Such nanoparticle system has been considered to eliminate or weaken the obstacles caused by TMTME by recognizing the antigens and receptors on cancer cell membranes[139]. Those nanoparticles were built to encapsulate and deliver miRNAs to specific lesion sites with enhanced solubility and efficacy of drugs, and reduced interaction with untargeted tissues[145-147]. Certain miRNA delivery systems were also developed for breast cancer treatment. For instance, in triple-negative breast cancer treatment, hyaluronic acid‑chitosan was built to deliver miRNA‑34a mimics[148], and RNA‑NPs(nanoparticles) decorated with EGFR‑targeting aptamer was for carry miRNA‑21 inhibitor[149]. A novel fabricated nano-complex, gold-nanoparticles (AuNPs) was developed to miR-206 for treating breast cancer[150]. However, application of delivery system also has disadvantages. For example, such approach will undoubtedly increase the cost, and the storage could become more difficult due to the crystallization process which might cause drug expulsion from the nanoparticles[151, 152] [153, 154]. Besides that, another promising technique is the embodiment of miRNA therapeutics into a biodegradable 3D matrix[155]. Such approach can reduce the side-effect by directly implantation into local lesion as part of a surgical intervention. Briefly, it has been proved that implanted 3D matrix can locally trigger a continuous, tissue-related and comparatively moderate release of miRNA-based curatives[156, 157]. In addition, ex-vivo miRNA-based therapeutics has also been regarded as a potential option which may provide more immediate clinical effect without going through complicated metabolic processes[155, 156]. Such technic has been shown to improve the effectiveness of adoptive immune cell transfer approaches, including potentiating adoptive T cell treatments, by regulating genes involved in T cell activation and fitness[155, 156]. Moreover, based on the potential curative effect possessed by certain miRNAs originated from medically relevant plant sources, a novel concept of oral administration of miRNA-based therapeutics that stem from plants has been provided as a potential solution[158]. However, many problems such as the bioavailability of miRNAs contained in plant food, and the regulatory capacity of plant miRNAs in mammalian cells, are still unclear[159, 160]. Therefore, massive investigations are required before any related therapeutic application can be implemented. Approaches such as topical, periocular and systemic corticosteroids administration, have also been developing for controlling related side effects[161-163]. Indeed, the selection of qualified and proper miRNA is most crucial. However, to achieve this goal, abundant of research are required, since it is a novel and innovative field.”

As for suggestions for alternatives to microRNAs for cancer therapy, we also added corresponding new context and discussion as suggested (Line 533, Page 13 - Line 555, Page 14), as showed below:

Additionally, alternatives to microRNAs, as sensitizer for ICIs based treatment, also require investment in research. Angiogenesis blockade, for example, has been found to exert better therapeutic effect when applied with immune checkpoint inhibitor in solid cancers[169]. Evidence also reported that appropriate angiogenesis inhibitor can alleviate immunosuppression and enhance immunity by pruning blood vessels that are pivotal for tumor progression, and blocking negative immune signals via decreasing the level of immune checkpoints, thereby increasing the anti-/pro-tumor immune subset ratio and alleviating hypoxia by normalizing tumor vasculature. Besides that, the efficacy of anti-angiogenic therapies can also be improved by ICBs via recruiting immune cell subtypes with angio-modulatory function[169]. Moreover, recent evidence has reported that histone deacetylases 2 (HDAC2) blockade can regulate the level of T lymphocyte subsets, inhibit tumor immune suppression and consequently enhance the efficacy of PD-1/PD-L1 inhibitor by regulating the progress of membranal PD-L1nuclear translocation and improving anti-cancer immunity[147, 170, 171]. Thus, HDAC2 inhibitors, as a novel type of anticancer drug, has also displayed its potential as a sensitizer for cancer immunotherapy[170]. CAXII (Carbonic anhydrase XII) inhibitor is also considered as a potential sensitizer which has been reported to reduce the immunosuppressive stress mediated by hypoxic/acidic metabolism, modulate the expression of CCL8 and affect the functions of monocytes and macrophages, thereby improving anti-tumor immunity and enhancing the therapeutic effect of PD-1 inhibitors in solid cancer[172-176]. Other approaches from alternative medicine such as human biofield therapy, have also been mentioned[177, 178]. Certainly, the combination strategy of multiple ICIs sensitizers may also provide novel perspective for future study of cancer treatment.”

Round 2

Reviewer 1 Report

 Authors should note that nanoparticles for microRNAs therapeutics would help in the delivery and even in specific tissue delivery, based on its design, however this would not eliminate the TMTME. 

Author Response

Response to Reviewer 1 Comments

Review 1 (round 2):

 Authors should note that nanoparticles for microRNAs therapeutics would help in the delivery and even in specific tissue delivery, based on its design, however this would not eliminate the TMTME. 

Answer

Thank the reviewer so much for his/her excellent comment. We totally agree with the reviewer that nanoparticles for microRNAs therapeutics would help in the delivery and even in specific tissue delivery. However, as described in article “MicroRNA Therapeutics in Cancer: Current Advances and Challenges” (doi:10.3390/cancers13112680): “nanoparticle-based delivery of miRNA aims to increase therapeutic efficacy, decrease the effective dose, and/or reduce the risk of systemic side effects”.

In other words, nanoparticle based delivery can send the therapeutic miRNAs to specific cancer cell/tissues therefore increasing the drug concentration in certain lesions, also decreasing systemic drug concentration and effective doses (doi: 10.1016/j.biomaterials.2019.02.016). Thus, as the doses, and also the systemic drug concentration were decreased, potentially, the amount of needless targets that supposed to be affected by delivered miRNAs could be reduced. Moreover, one of AuNPs mentioned in our manuscript has been proved to accelerate body clearance of off‐target drug. Therefore, the “too many targets for miRNA effect (TMTME)” may be weakened by applying nanoparticle. In addition, our previous study also proved the systemic toxicity-reducing effect of certain nanoparticle (LNP-DP1) by efficient delivery (doi: 10.1016/j.omto.2020.08.017).

Furthermore, even though we discussed the nanoparticle as a potential approach for weakening TMTME, more research is apparently needed in such field. Therefore, we further revised our relevant description and shown as followed (from Line 480 in Page 12 to Line 494 in Page 12):

“One of the potential solutions is the application of safe and targeted drug delivery system which is a pattern of specifically designed carriers, largely based on nanomedicine[145, 146]. Such nanoparticle system has been considered to weaken the obstacles caused by TMTME by decreasing systemic drug concentration and effective doses[139]. Those nanoparticles were built to encapsulate and deliver miRNAs to specific lesion sites with enhanced solubility and efficacy of drugs, and reduced interaction with untargeted tissues[145-147]. Thus, potentially, the amount of needless targets that supposed to be affected by delivered miRNAs could also be reduced [145-147]. Certain miRNA delivery systems were also developed for breast cancer treatment. For instance, in triple-negative breast cancer treatment, hyaluronic acid‑chitosan was built to deliver miRNA‑34a mimics[148], and RNA‑NPs(nanoparticles) decorated with EGFR‑targeting aptamer was for carry miRNA‑21 inhibitor[149]. A novel fabricated nano-complex, gold-nanoparticles (AuNPs) was developed to miR-206 for treating breast cancer[150]. However, as potential approach for dealing with the TMTME, more research and still required to further evaluate the relevant effects of nanoparticle system.”
